# A Micromechanical Analysis to the Viscoplastic Behavior of Sintered Silver Joints under Shear Loading

**DOI:** 10.3390/ma16124472

**Published:** 2023-06-19

**Authors:** Kun Ma, Xun Liu, Yameng Sun, Yifan Song, Zheng Feng, Yang Zhou, Sheng Liu

**Affiliations:** 1Laboratory for Electronic Manufacturing and Packaging Integration, the Institute of Technological Sciences, Wuhan University, Wuhan 430070, China; mk_175@whu.edu.cn (K.M.); xun.liu@whu.edu.cn (X.L.); sunyameng@whu.edu.cn (Y.S.); 2School of Mechanical Science and Engineering, Huazhong University of Science and Technology, Wuhan 430070, China; syfmvp@163.com; 3Hefei Archimedes Electronic Technology Co., Ltd., Hefei 230094, China; fzheng_kite@163.com; 4School of Power and Mechanical Engineering, Wuhan University, Wuhan 430070, China

**Keywords:** crystal plasticity, fatigue life, finite element modeling, multi-scale silver paste, shear stress–strain curve

## Abstract

Ag paste has been recognized as a promising substitute for Sn/Pb solder in SiC or GaN power electronic devices, owing to its ability to withstand high temperatures and facilitate low-temperature packing. The reliability of these high-power circuits is greatly influenced by the mechanical properties of sintered Ag paste. However, there exist substantial voids inside the sintered silver layer after sintering, and the conventional macroscopic constitutive models have certain limitation to describe the shear stress–strain relationship of sintered silver materials. To analyze the void evolution and microstructure of sintered silver, Ag composite pastes composed of micron flake silver and nano-silver particles were prepared. The mechanical behaviors were studied at different temperatures (0–125 °C) and strain rates (1 × 10^−4^–1 × 10^−2^) for Ag composite pastes. The crystal plastic finite element method (CPFEM) was developed to describe the microstructure evolution and shear behaviors of sintered silver at varied strain rates and ambient temperatures. The model parameters were obtained by fitting experimental shear test data to a representative volume element (RVE) model built on representative volume elements, also known as Voronoi tessellations. The numerical predictions were compared with the experimental data, which showed that the introduced crystal plasticity constitutive model can describe the shear constitutive behavior of a sintered silver specimen with reasonable accuracy.

## 1. Introduction

The demands of sophisticated high-power electronic devices in aerospace and new energy vehicles have resulted in the working environment and load conditions of next-generation chip packaging becoming increasingly complex. These extreme operating conditions pose significant challenges to traditional interconnect materials [1,2,3]. Nano-silver paste, due to its exceptional properties such as a high melting temperature (961.8 °C), impressive thermal conductivity (200–300 W/(m·K)), and long-term durability, has emerged as a potential substitute for Sn/Pb solders in SiC or GaN power electronics [4,5,6,7,8,9]. For instance, power dissipation and ambient temperature alterations during the packaging, testing, and service processes subject electronic packaging and its components to a cyclical temperature effect [10]. Given that the chip, substrate, and sintered silver layer each possess distinct coefficients of thermal expansion (CTE), alternating shear stress is produced at the interconnect junction. The subsequent accumulation of plastic strain in the sintered silver layer amplifies the accumulation of damage, potentially leading to failure under the impact of thermal stress [11,12]. Consequently, employing experimental and simulation methods to investigate the mechanical properties of sintered silver is of paramount importance.

In order to elucidate the mechanical properties of nano-silver, researchers have conducted numerous experiments and theoretical analyses. Ide et al. [13] explored the mechanical properties of the sintered layer and the interfacial connectivity of Cu-Cu, employing nano-silver solder, and considering the impact of sintering pressure, temperature, and duration. Li et al. [14] scrutinized the high-temperature ratcheting behavior of lap shear samples sintered with nano-silver paste. Furthermore, Yu et al. [15] performed a series of tensile tests to analyze the mechanical performance of nano-silver paste under various strain rates and ambient temperatures. He et al. [16] utilized mechanical tests and Weibull simulation to investigate the fracture behavior of sintered porous nano-silver joints of diverse sizes.

In addition to these experimental pursuits, substantial efforts have been directed toward establishing the constitutive relationship of nano-silver paste. The Anand constitutive model is widely acknowledged as a potent tool for predicting the stress–strain relationship of soldered materials at diverse temperature settings in microelectronics [15,17,18,19]. However, the Anand model, being a phenomenological model, primarily focuses on the macroscopic material properties of silver paste.

Given the trend toward miniaturization in electronic packaging, the influence of microstructures on the anisotropic mechanical properties of interconnected materials cannot be overlooked [20]. The use of sintered nano-silver as a packaging material hinges heavily on its mechanical properties, but the Anand model does not account for the microstructure features. Consequently, it poses challenges to accurately determine the mechanical characteristics and microstructure evolution of the sintered silver layer. Additionally, there are relatively few studies on the constitutive relationship pertaining to the shear deformation of sintered nano-silver paste [20,21].

Given these insights, it is necessary to propose a model to study the evolution of microstructure and shear behavior at the grain level. With the advancement of crystal plasticity and the dislocation slip theory, constitutive models of crystal plasticity have been formulated to evaluate several factors that influence grain deformation, including crystal orientation, crystal structure, and the effects of slip or twinning mechanisms [22,23]. The crystal plasticity constitutive model presents apparent advantages over the traditional plastic constitutive model by recognizing microstructure features and providing superior modeling for the characterization of plastic strain localization [24]. Numerical models grounded in the crystal plasticity constitutive theory are employed to examine the micromechanical behavior of crystal materials, particularly the high-temperature fatigue properties of these materials [25,26,27,28]. Gong et al. [29] employed the crystal viscoplastic model to investigate the anisotropic performance of SnAgCu solder junctions in single crystal, twin crystal, and polycrystalline scenarios. Meanwhile, Zamiri et al. [30] developed a crystal plasticity finite element model to study the impact of crystal orientation, as well as solder joint size and shape, on the reliability of lead-free solder joints. Additionally, Darbandi et al. [31] optimized the crystal plasticity model parameters for lead-free tin-based solders, enabling predictions of slip mechanisms that contribute to damage accumulation in lead-free solder connections. Zhang et al. [32] proposed an anisotropic constitutive model coupled with damage to characterize the deformation mechanism of tin-rich solder.

The majority of the existing research has primarily focused on tin-based solder, creating a gap in the literature concerning the crystal–plastic finite element model parameters that can accurately depict the microstructure evolution and shear behaviors of sintered silver solder paste. Therefore, there is a pressing need to introduce a crystal plasticity finite element model for sintered silver paste and to determine its associated crystal plasticity parameters. The present study employs a developed model to simulate the shear deformation behavior of silver joints at varying strain rates and temperatures. A hybrid solder composed of nano and micron-sized silver particles was fabricated, with its sintering performance meticulously examined. Numerical and experimental shear tests were conducted on the sintered silver paste. A unique crystal plasticity model that couples kinematic hardening and fine-grain strengthening was developed and implemented in ABAQUS using a user-defined material (UMAT) subroutine. A representative volume element (RVE) model was constructed based on Voronoi tessellations, and model parameters were derived by fitting the experimental shear test results. Subsequently, a convergence analysis of grain orientation and grid size was performed. The resultant model effectively simulates the shear deformation behavior of silver joints under various strain rates and temperatures, providing significant insights into the mechanical properties and behavior of sintered silver paste.

## 2. Constitutive Model Theory

### 2.1. Finite Element Theoretical Model of Crystal Plasticity

The theory of crystal plasticity elucidates the relationship between material microstructure and macromechanical response, probing material deformation and the constitutive theory from the perspective of the material’s inherent deformation mechanisms. In comparison with traditional constitutive models, it offers the distinct advantage of capturing the essence of material plastic deformation.

To portray the impact of a crystal’s internal properties on the stress–strain characteristics of materials during the elastic–plastic process, an array of crystal plasticity constitutive models have been devised. The crystal structure of Ag composite pastes is face-centered cubic (FCC), with the slip system utilized in this study being (111) <110>. In the process of characterizing crystal deformation properties, the overall deformation gradient in the Cartesian coordinate system is bifurcated into elastic and inelastic components, as depicted in Equation (1) [33]:(1)$$F=F_{e}F_{in}$$
where $F_{e}$ represents elastic deformation gradient and rigid rotation, and $F_{in}$ stands for the inelastic deformation gradient. The inelastic shear rate of the slip framework based on kinematic dislocation is proportional to the velocity gradient L, which is specified by the deformation gradient F. The corresponding equation is shown in Equation (2):(2)$$L=\dot{F}\cdot F^{-1}$$

The symmetric deformation tensor D and the antisymmetric rotation tensor W, which can be further divided into elastic part $\left(D_{e},W_{e}\right)$ and inelastic part, respectively $\left(D_{in},W_{in}\right)$, can be combined to generate the velocity gradient L:(3)$$\begin{array}{c} L=D+W\\ D=\frac{1}{2}\left(L+L^{T}\right)=D_{e}+D_{in}\\ W=\frac{1}{2}\left(L-L^{T}\right)=W_{e}+W_{in} \end{array}$$

The corresponding inelastic part $\left(D_{in},W_{in}\right)$ is shown in Equation (4):(4)$$\begin{array}{c} D_{in}\;=\sum_{a=1}^{n}{\dot{\gamma }}^{a}u^{a}=\sum_{a=1}^{n}{\dot{\gamma }}^{a}\frac{s_{e}^{a}⊗m_{e}^{a}+m_{e}^{a}⊗s_{e}^{a}}{2}\\ W_{in}\;=\sum_{a=1}^{n}{\dot{\gamma }}^{a}\omega^{a}=\sum_{a=1}^{n}{\dot{\gamma }}^{a}\frac{s_{e}^{a}⊗m_{e}^{a}+m_{e}^{a}⊗s_{e}^{a}}{2} \end{array}$$
where ${\dot{\gamma }}^{a}$ stands for the inelastic shear rate of the slip system a,n denotes the total number of the slip systems, $u^{a}$ and $\omega^{a}$ represent the symmetric and skew-symmetric slip system tensors, respectively. The dislocation slip plane and normal directions are $s_{e}^{a}$ and $m_{e}^{a}$, respectively. Here, the slip systems applied matched those of the task. The following equations can be used to define the variation of the slip plane $s_{e}^{a}$ and normal $m_{e}^{a}$ during inelastic deformation: (5)$$\begin{array}{cc} s_{e}^{a} & =F_{e}F_{in}s^{a}\\ m_{e}^{a} & =m^{a}F_{in}^{-1}F_{e}^{-1} \end{array}$$

The relation between Jaumann rate of Cauchy stress $\sigma^{\nabla_{e}}$ and the symmetric rate of lattice stretching $D_{e}$ can be delineated using the framework of the elastic constitutive equation proposed by Hill and Rice [34], as shown in Equation (6):(6)$$\sigma^{\nabla_{e}}+\sigma \left(I:D_{e}\right)=L_{e}:D_{e}$$
where $L_{e}$ is the elastic tensor and I is the fourth-order identical tensor. Equation (7) illustrates the Jaumann rate of Cauchy stress $\sigma^{\nabla }$:(7)$$\sigma^{\nabla_{e}}=\sigma^{\nabla }-W_{e}\cdot \sigma +\sigma \cdot W_{e}$$
where $\sigma^{\nabla_{e}}$ stands for the corotational stress rate on axes that rotate with the material and is depicted in Equation (8).
(8)$$\sigma^{\nabla }=\dot{\sigma }-W\cdot \sigma +\sigma \cdot W$$

### 2.2. Inelastic Shear Rate Function

Equation (9) is utilized to explain the inelastic shear strain rate ${\dot{\gamma }}^{a}$ and to predict the relationship between temperature and strain rate.
(9)$${\dot{\gamma }}^{a}={\dot{\gamma }}_{0}e^{-\frac{Q}{KT}}{\left(\sinh\left(\left|\frac{\tau^{a}-\chi^{a}}{g^{a}}\right|\right)\right)}^{n}sign\left(\tau^{a}-\chi^{a}\right)$$

n is the exponent accounting for rate dependence, ${\dot{\gamma }}^{a}$ is the reference shear rate, Q is the activation energy, T is the absolute temperature, K is the Boltzmann constant, $\tau^{a}$ is the critical resolved stress, and $\chi^{a}$ is the back stress.

The critical resolved stress is shown in Equation (10):(10)$$\tau_{a}=m_{e}^{a}\cdot \sigma \cdot s_{e}^{a}$$

According to the nonlinear kinematic hardening law of the Armstrong–Frederick equation [35], the back stress $\chi^{a}$ is defined as shown in Equation (11):(11)$${\dot{\chi }}^{a}=h_{1}{\dot{\chi }}^{a}-h_{2}\left|{\dot{\chi }}^{a}\right|\chi^{a}$$

$h_{1}$ and $h_{2}$ are material constants and ${\dot{\chi }}^{a}$ is the rate at which back stress is changing. The hardening strength $g^{a}$, which reflects the slip system’s strain hardening behavior, is illustrated in Equation (12): (12)$$g^{a}=\overset{n}{\underset{a=1}{\sum }}h_{a\beta }{\dot{\gamma }}^{\beta }$$

$h_{a\beta }$ stands for a slip-hardening modulus matrix and specifies the slip system’s self-hardening modulus for a=β. $h_{a\beta }$ is also known as the latent hardening modulus in instance (a≠β), indicating that another slip mechanism is to blame for the hardening behavior.
(13)$$h_{a\beta }=\left\{\begin{array}{l} h(\gamma )=h_{0}{\mathrm{sech}}^{2}\left|\frac{h_{0}\gamma_{tol}}{\tau_{s}-\tau_{0}}\right|(a=\beta )\\ qh(\gamma )(a\neq \beta ) \end{array}\right.$$

$h_{0}$ is the ratio of the latent hardening parameter to the self-hardening behavior of crystalline, and its value typically ranges from 1 to 1.4. H is the initial hardening rate, $\tau_{s}$ stands for the saturation stress, $\tau_{0}$ represents the initial yield stress, and q represents the initial yield stress. q was set to 1 to make the simulation simpler [34]. $\gamma_{tol}$ represents the total shear strain experienced by all slip systems.
(14)$$\gamma_{tol}=\overset{N_{v}}{\underset{a=1}{\sum }}\left|\gamma^{a}\right|$$

By creating the UMAT subroutine in Fortran, these relationships were added to the commercial finite element software ABAQUS 6.14. This constitutive model’s solution was achieved by using forward Euler integration. The crystal plasticity model’s code structure was based on Zhang et al.’s work [32].

## 3. Experimental Section

### 3.1. Preparation of the Mixed Paste

The silver nanoparticles utilized in this study, which consist of spherical nano-scale particles smaller than 50 nm, were supplied by the Shenzhen Institute of Advanced Technology (Shenzhen, China) [35,36,37,38]. Flake micron silver was selected as the supporting structure in the mixed silver paste. This flake silver powder, in theory, possesses a higher theoretical bulk density and specific surface area compared to spherical silver powder, suggesting a stronger sintering driving force and a likelihood of achieving a denser sintering structure. The micronized Ag flakes used in this study were PAg-S8, produced by Buwei (Shanghai, China) Co., Ltd. Figure 1 illustrates the morphology of the prepared silver nanoparticles and flake micro-silver. The silver nanoparticles are uniformly spherical, with consistent morphology and size. The dispersion between the particles indicates that the prepared silver nanoparticles have good dispersibility. The average particle size of the silver nanoparticles is 50 nm, with a distribution range of 40–80 nm. Figure 1b depicts the morphology of the flake micron silver powder used in the experiment. The mean particle size of the silver powders is 5 μm, and the sheet thickness is approximately 250 nm.

Hybrid pastes were prepared using the following steps: Firstly, the purchased flake micron silver was pre-treated to remove long-chain fatty acids on the surface. The nano-silver powder and the pre-treated micron silver powder were then combined in a 2:3 ratio, to which the prepared organic carrier was added. The mass ratio of the organic carrier to the mixed powder was 1:4. Finally, a planetary gravity mixer was employed to blend the weighed organic carrier and the mixed powder at a speed of 1000 rpm for 80 s to yield the mixed paste.

### 3.2. Preparation of Shear Specimens

To investigate the failure patterns of sintered nano-silver materials in electronic packaging components, lap shear joint samples were designed to emulate real solder joints. Shear tests were conducted using the MTS C43.104E universal testing instrument (MTS, Shanghai, China) at temperatures ranging from 0 °C to 125 °C and strain rates of 1 × 10^−4^ to 1 × 10^−2^. Figure 2a presents the schematic for the thermomechanical coupling test apparatus. A schematic and actual image of the lap shear specimen is shown in Figure 2b and 2c, respectively.

The shear test specimens consist of two symmetrical silver-plated copper plates and a sintered nano-silver layer. The nano-silver layer’s dimensions are 3 mm in length, 3 mm in width, and 0.1 mm in thickness, yielding a sintered layer area of 9 mm^2^. The bonding process was carried out at 175 °C for 15 min with a pressure of 10 MPa, following a preheating stage at 75 °C for 10 min.

## 4. Determination of Parameters

### 4.1. Finite Element Model

From a mesoscopic viewpoint, the use of finite element analysis, grounded in the RVE model, has been widely adopted to probe the macroscopic mechanical behavior of materials [22,39].

Figure 3 exhibits the pole figure that represents the grain orientation. Figure 4a presents a Voronoi tessellation polygon, created from randomly generated seed nodes, to establish a model comprising 200 grains. The RVE model measures 400 μm on each side and is discretized into eight-node linear bricks (C3D8) with a mesh size of 40 μm.

As shown in Figure 4b, a reference point P was positioned above the top face to obtain the force through kinematic coupling when applying a z-displacement to point P. To ensure alignment between the simulation results and the experimental findings, periodic boundary conditions were incorporated into the RVE model.

### 4.2. Parameters Identification

It is essential to obtain the parameters of the proposed constitutive model. The material parameters in the proposed model include Young’s modulus ($L_{e}$), thermal activation parameter (Q), flow parameters (${\dot{\gamma_{0}}}^{\alpha }$,$m,g_{0}$), hardening parameters ($h_{0},h_{s},\tau_{s},\tau_{0},q$), and back-stress parameters ($h_{1},h_{2}$). For a cubic crystal, only three independent Young’s modulus ($L_{11}$,$L_{12}$,$L_{44}$) are needed based on the symmetry of its FCC crystal structure.

In order to determine the parameters of the crystal plasticity finite element model, the average absolute error is used as the optimization function. Here, the objective function is shown in Equation (15):(15)$$O=\overset{n}{\underset{i=1}{\sum }}\left(\left|\frac{y\left(x_{i}\right)-f\left(x_{i}\right)}{f\left(x_{i}\right)}\right|\right)$$
where $y\left(x_{i}\right)$ and $f\left(x_{i}\right)$ are the simulated and measured results, respectively, and n is the total number of the experimental results. The ideal constitutive parameters were found through a procedure of trial and error, as indicated in Table 1. The simulation findings with specific settings are in good agreement with the experimental data, as shown in Figure 5.

### 4.3. Convergence Study

The accuracy of the simulation results is influenced by the mesh size and the number of grains. To ensure the reliability of the obtained material parameters, it is essential to establish multiple simulation models. Previous research has suggested that grain orientation effectively encapsulates the impact of grain number. As a result, in this study, three random orientations (Orientation No. 1, Orientation No. 2, Orientation No. 3) were assigned to the RVE model to assess convergence with respect to grain number.

Figure 6 illustrates the stress–strain curves for three different grain orientations, with each orientation represented by a distinct color. When the material transitions into plastic deformation, the stress difference between these three models is less than 3%. Figure 7 displays the distributions of stress, with the maximum stress of the three models found in different locations. This suggests that, while grain orientation significantly influences the distribution of stress, the overall stress–strain response of the model is not significantly affected by grain orientation.

Similarly, in order to study the convergence of element size, three different mesh sizes (20.0 μm, 15 μm, and 10.0 μm) were used in the RVE model with the same grain orientation.

The stress–strain curves corresponding to three different mesh sizes are depicted in Figure 8. When the material enters plastic deformation, the impact of mesh size on the stress–strain curves is minimal, with the stress difference between the various models being less than 1%. Figure 9 presents the stress distributions for each of the three models. All three models exhibit the same distribution, and the stress difference between these models is less than 1%. This indicates that a mesh size of 20 μm is sufficient to satisfy the simulation accuracy requirements.

From the above analysis, it is concluded that a 200-grain count and a 20 μm mesh size used in the RVE model for parameter determination are appropriate.

## 5. Results and Discussion

### 5.1. Fracture Surface and Void Distribution

Figure 10 presents the SEM analysis of the fracture microstructures of the sintered samples at varying shear temperatures. When the sintered structure of the interconnect joint undergoes shear stress, the voids within the joint structure and the silver sintered structure collaboratively deform under it. As the shear stress escalates, the sintered structure between voids gradually contracts and eventually fractures. The cross-sectional microstructure appears apical and vertebral. The cavities within the joint tissue evolve into dimples in the fracture tissue. 

As Figure 10 illustrates, with the increase in shear temperature, the cross-sectional microstructure of the sintered nano-silver layer becomes coarser to some extent. This phenomenon can be attributed to the heightened temperature which intensifies lattice vibrations. Consequently, the resistance to atomic movement decreases and the diffusion speed is accelerated. Fusion between silver nanoparticles occurs, and large grains subsume smaller ones, leading to an evident coarsening of the microstructure. When the temperature was 0 °C, the fracture surface had few sintered necks, indicating that the plastic deformation did not occur. When the temperature rose to 125 °C, the density of the bondline increased, sintered necks further grew up, and the grain was significantly larger than that of sintering at 0 °C. Therefore, the high service temperature resulted in thicker sintered necks. The void ratio of the nano-silver layer significantly impacts the joint’s shear strength and thermal properties. Image J 2.0 software was employed to process the scanning electron microscope images of nano-silver joints and compute the porosity. By adjusting contrast, brightness, and threshold, the ratio of the area within a certain threshold range to the total area was calculated to obtain the corresponding porosity. The black regions depicted in Figure 11 represent voids or defects in the sintered nano-silver samples. Figure 11 shows the void distribution of sintered nano-silver joints at different shear test temperatures. When the shear test temperature is 0 °C, the porosity of the sintered silver layer is 13.94%. This porosity decreases to 7.72% when the shear test temperature escalates to 125 °C. It can be observed that the void ratio of the joint decreases with an increase in shear temperatures.

### 5.2. Shear Stress Responses with Different Strain Rates and Temperatures

The shear strengths of sintered nano-silver joints, as obtained from the shear tests, are summarized in Figure 12. The shear strengths of the specimens demonstrate a linear decrease with the rise in shear temperature. The sintered nano-silver joints possess a shear strength range of 11.01–45.4954 MPa, given a temperature range of 0–125 °C and loading rates of 1 × 10^−4^–1 × 10^−2^. Figure 12b provides a comprehensive summary of the effects of strain rates on the obtained shear strength of the sintered joints. The shear strength of the joints typically increases with strain rates, a phenomenon that can be attributed to the strengthening effect of the loading rates.

### 5.3. Simulation Results of Nano-Silver

A series of shear test simulations for nano-silver joints were conducted to investigate the parameters of the proposed model, as depicted in Figure 13. The simulation results are represented by dashed lines, while the solid lines denote the experimental data. The CPFEM model parameters used for nano-silver joints are presented in Table 1. The developed model simulates the shear deformation of nano-silver joints across various strain rates and temperatures. 

Upon calibration, the stress difference between the simulation results and the experimental data is less than 10%, indicating that the developed model can fit the experimental data with a satisfactory level of accuracy.

## 6. Conclusions

In this work, a mixed solder composed of micron flake silver and nano-silver particles was developed, and the sintering performance of the lap joints using this composite solder was systematically examined. 

Shear tests were conducted on the nano-silver joints at five distinct shear strain rates (ranging from 0.01 s^−1^ to 0.0001 s^−1^) in conjunction with four different temperatures (from 0 °C to 125 °C) to ascertain the proposed model parameters. The shear strength of the sintered nano-silver joints varied between 11.01 and 45.4954 MPa, with their porosity ranging from 13.94% to 7.72%.

A viscoplastic constitutive model that encapsulates the thermomechanical characteristics of the nano-silver solder was proposed and incorporated into the user subroutine UMAT of the commercial finite element software ABAQUS 6.14, based on the theory of crystal plasticity. An RVE simulation model was employed to fit the shear curves of the materials at diverse temperatures and strain rates, allowing us to determine the material parameters of the constitutive model.

The Voronoi grain generation method was utilized to randomly generate 200 grains with random orientations in the RVE model, followed by a convergence analysis of grain orientation and grid size. The model simulates the shear deformation of nano-silver junctions at an array of strain rates and temperatures. The results affirm that the model can successfully align with the experimental data. The material behavior of sintered silver solder can be accurately depicted using the crystal plasticity constitutive model.

## Figures and Tables

**Figure 1 materials-16-04472-f001:**
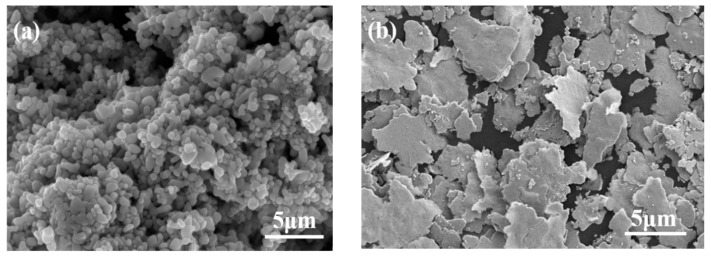
SEM micrographs of (**a**) Ag nanoparticle, (**b**) Ag flake.

**Figure 2 materials-16-04472-f002:**
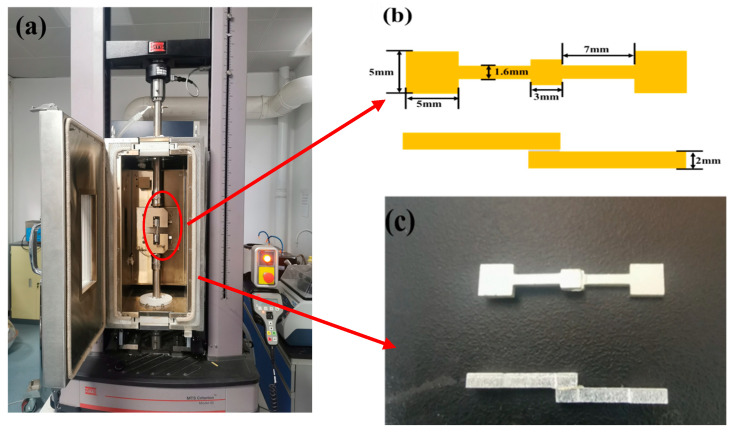
Experimental device and specimen figure. (**a**) Sample tester; (**b**) Schematic of specimen; (**c**) Actual specimen.

**Figure 3 materials-16-04472-f003:**
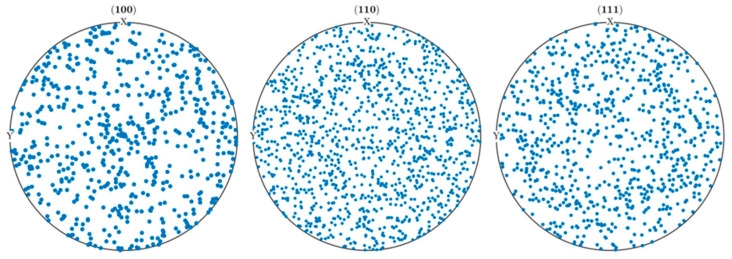
Pole figures for RVE simulations model. RVE: representative volume element.

**Figure 4 materials-16-04472-f004:**
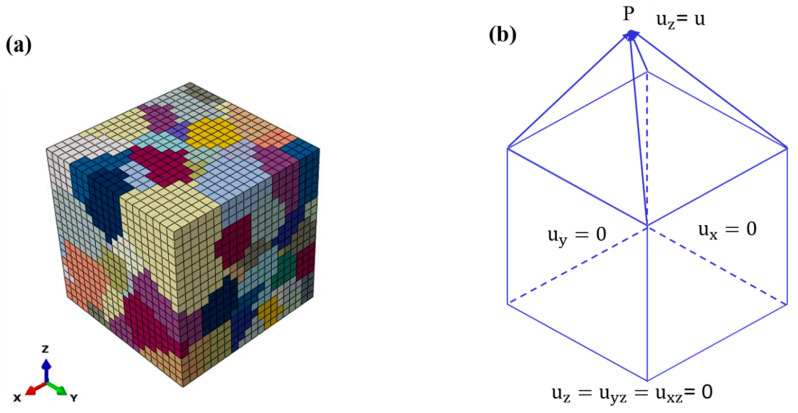
(**a**) RVE model for model parameter determination; (**b**) boundary conditions imposed on RVE model.

**Figure 5 materials-16-04472-f005:**
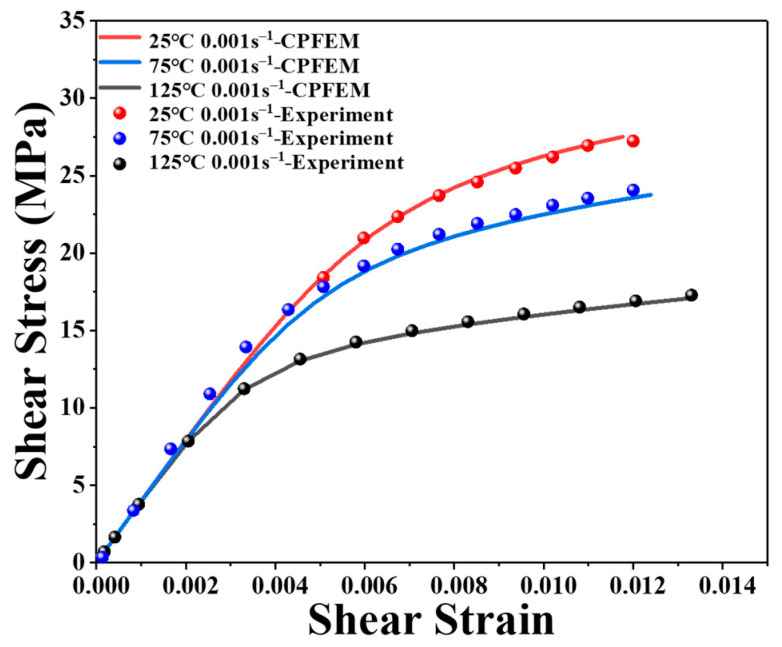
Comparison of experimental data and simulation results.

**Figure 6 materials-16-04472-f006:**
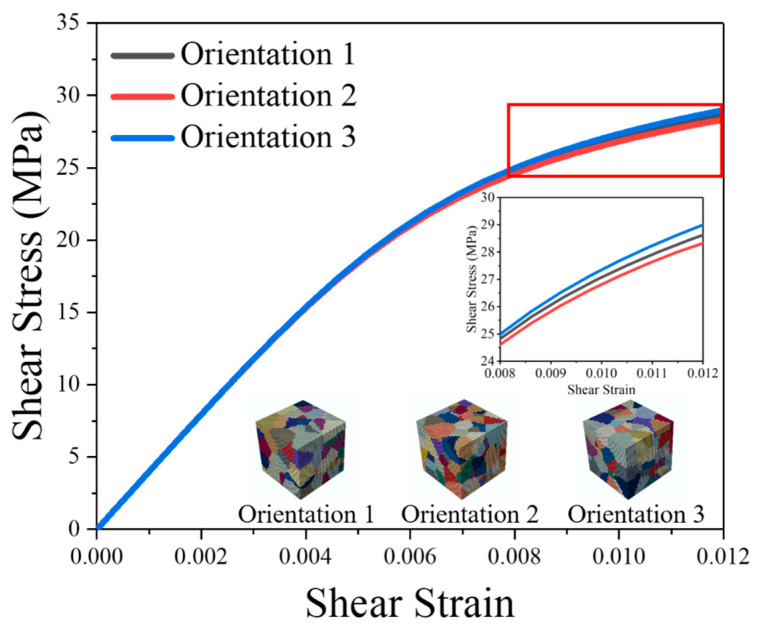
Stress–strain curves of RVE models with different orientations.

**Figure 7 materials-16-04472-f007:**
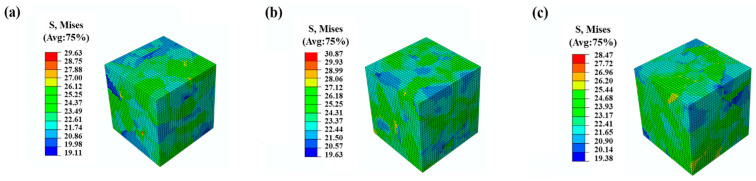
Distributions of Von Mises stress of the RVE models with three orientations under the strain of 0.1%. (**a**) Orientation No. 1; (**b**) Orientation No. 2; (**c**) Orientation No. 3.

**Figure 8 materials-16-04472-f008:**
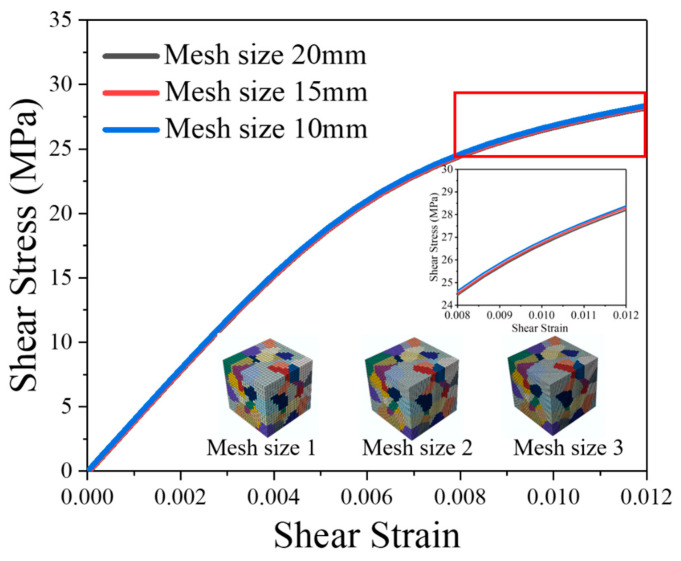
Stress–strain curves of RVE models with different mesh sizes.

**Figure 9 materials-16-04472-f009:**
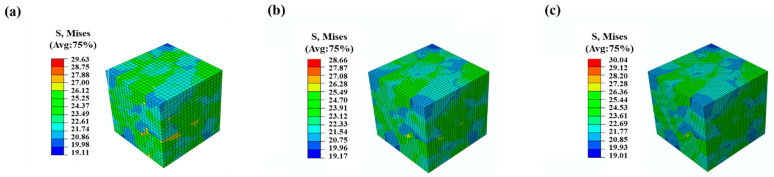
Distributions of Von Mises stress of the RVE models with three mesh sizes under the strain of 0.1%. (**a**) 20.0 μm; (**b**) 15 μm; (**c**) 10.0 μm.

**Figure 10 materials-16-04472-f010:**
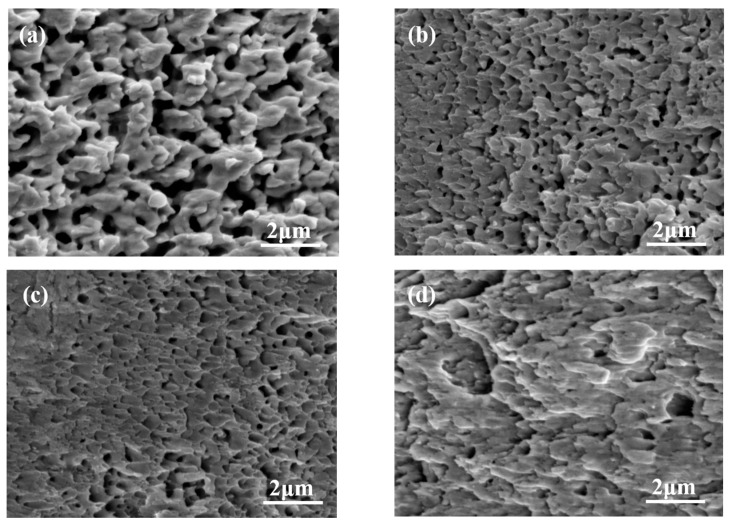
Microstructures of fracture surface of sintered nano-silver layer observed by SEM: (**a**) 0 °C, (**b**) 25 °C, (**c**) 75 °C, (**d**) 125 °C.

**Figure 11 materials-16-04472-f011:**
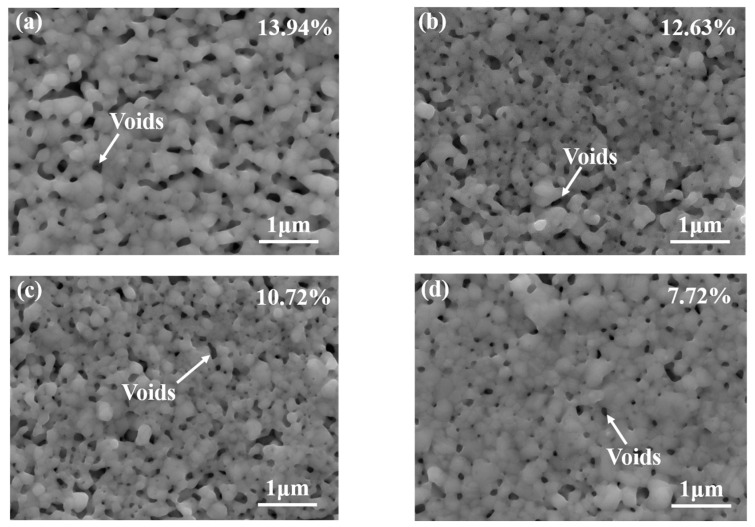
Porosity of nano-silver layer observed by SEM: (**a**) 0 °C, (**b**) 25 °C, (**c**) 75 °C, (**d**) 125 °C.

**Figure 12 materials-16-04472-f012:**
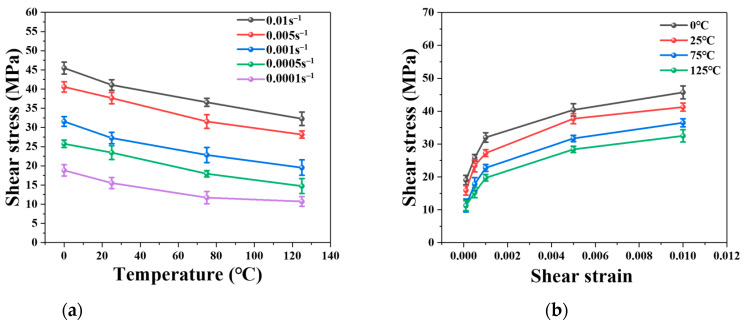
Comparison of shear strength: (**a**) Shear temperature, (**b**) Strain rate.

**Figure 13 materials-16-04472-f013:**
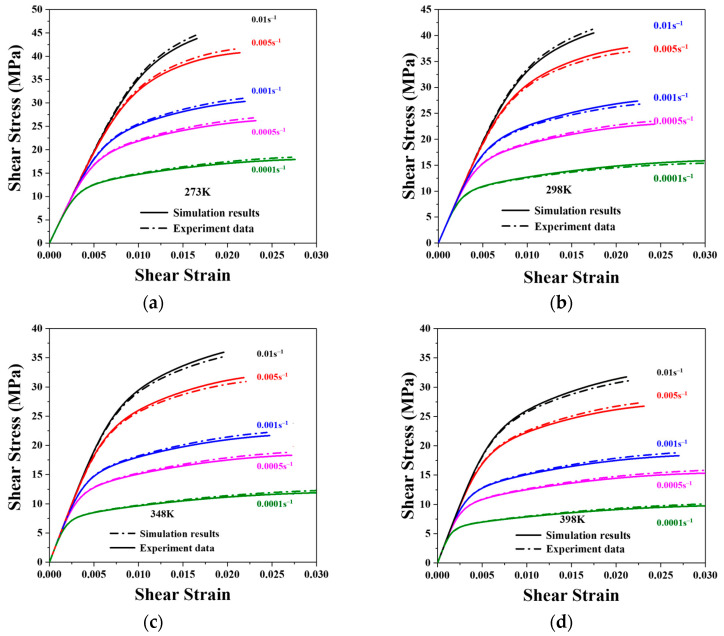
Comparison between the proposed model and experimental data of nano-silver joints. (**a**) 273 K; (**b**) 298 K; (**c**) 348 K; (**d**) 398 K.

**Table 1 materials-16-04472-t001:** The CPFEM parameters of the sintered nano-silver.

Parameter Number	CPFEM Parameter	Unit	Nomenclature	Value
1	$L_{11}$	GPa	Elastic constants	5.384
2	$L_{12}$	GPa		2.307
3	$L_{44}$	GPa		1.538
4	Q	KJ mol^−1^	Activation energy	14,000
5	${\dot{\gamma_{0}}}^{\alpha }$	-	Inelastic slip rate	0.54
6	n	-	Dislocation plasticity strain-rate exponent	3
7	$\tau_{s}$	MPa	Saturation critical resolved stress	18
8	$\tau_{0}$	MPa	Initial critical resolved stress	11.5
9	$h_{1}$	MPa	Material constants	20
10	$h_{2}$	MPa	Material constants	15
11	$h_{0}$	MPa	Initial hardening rate	100
12	q	-	Ration between self-hardening and latent hardening	1

## Data Availability

The data that support the findings of this study are available from the corresponding author upon reasonable request.

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
