# Peer review of "A Micromechanical Analysis to the Viscoplastic Behavior of Sintered Silver Joints under Shear Loading"

_materials, 2023, doi:10.3390/ma16124472_

Round 1

Reviewer 1 Report

Materials 2457895

The paper describes a viscoplastic constitutive model to capture the thermomechanical behaviour of a mixed solder composed of micron flake silver and nano silver particles. The parameters of the constitutive model have been identified using shear tests at five shear strain rates and four different temperatures. The agreement between simulation results and experimental results has been very good. The constitutive model has been implemented in the commercial software Abaqus.

The paper research methods are robust, the obtained results are valuable and the paper is well organized and clearly written. Therefore, I suggest that the manuscript is accepted for publication in its current form. Few minor corrections are suggested:

11)     Abstract: “…experimental shear test data to an representative volume…” should be “…experimental shear test data to a representative volume…”

22)    Introduction: I suggest to simplify the following wordy sentence: “In light of these insights, it is compelling to propose an intuitive and pragmatic framework to examine…”

33)    “4.2 Parameters Obtaining”, I suggest “4.2 Parameters identification”

Author Response

We highly appreciate your helpful suggestions and comments on our manuscript entitled “A Micromechanical Analysis to the Viscoplastic Behavior of Sintered Silver Joints under Shear Loading” (materials-2457895). We have revised our manuscript following your suggestions carefully. The highlighted revisions are described in the attachment.  

Reviewer 2 Report

1. Could we increase the number of tests, at least one more, to ensure the validation?

2.  Voids are non-homogenously distributed. Is there any measurement based on temperature for void size?   

3. What is the maximum temperature that can be applied for the paste, as per Figure 11, increasing the temperature reduction in the size of the voids? 

4. Generally, the melting temperature of nanoparticles varies between 112 to 153 °C, in case the test can be extended to 135 or 140 degrees to understand the binding ability. 

5. Could you also provide the melting point of the paste separately because that gives a better understanding of the binding ability of the article? 

Moderate correction is required 

Author Response

We highly appreciate your helpful suggestions and comments on our manuscript entitled “A Micromechanical Analysis to the Viscoplastic Behavior of Sintered Silver Joints under Shear Loading” (materials-2457895). The constructive suggestions are truly valuable for further improving our manuscript. We have revised our manuscript following your suggestions carefully. The highlighted revisions are described in the following.

Reviewer 3 Report

REVIEW

on article

A Micromechanical Analysis to the Viscoplastic Behavior of Sintered Silver Joints under Shear Loading

Kun Ma, Xun Liu, Yameng Sun, Yifan Song, Zheng Feng, Yang Zhou and Sheng Liu

SUMMARY

One of the topical areas of modern electronics is to increase the reliability of power semiconductor devices through the use of technology for connecting their elements using silver-containing pastes.

The mechanical strength of silver-containing structures is largely determined by the adhesive properties of the sintered layer.

The adhesive characteristics of the sintered layers of silver pastes depend significantly on the technological conditions of the sintering process. To study this effect, it is necessary to use optimal methods for measuring the adhesive strength of sintered layers, which ensure the reliability of measurements.

The article submitted for review is devoted to the study of the viscoplastic behavior of sintered silver joints under shear loading. The authors prepared composite Ag pastes consisting of micron silver flakes and silver nanoparticles, and systematically investigated the sintering characteristics of mixed solder. The authors have studied the microstructure and mechanical shear characteristics of sintered silver at various strain rates and ambient temperatures.

In general, the authors have done a lot of work and obtained a model based on the crystal plastic finite element method.

The references list comprises 36 sources.

Thus, the study is dedicated to a relevant scientific problem. At the same time, there are several comments that need to be corrected. Remarks are given below.

COMMENTS

1.        I recommend the authors revise the Abstract. There is no formulation of the research gap and scientific problem. Why this study is important? What is the scientific problem that the authors were solving? Editors strongly recommended the following structure of the Abstract: 1) Background: Place the question addressed in a broad context and highlight the purpose of the study; 2) Methods: Describe briefly the main methods or treatments applied. Include any relevant preregistration numbers, and species and strains of any animals used. 3) Results: Summarize the article's main findings; and 4) Conclusion: Indicate the main conclusions or interpretations.

2.        The introduction is well written. However, it is difficult to highlight the research gap and scientific problem that the authors are solving from the above review of the literature. I encourage the authors to highlight the scientific problem they are solving based on the gaps identified in the review.

3.        I recommend that the authors, in the last paragraph of the Introduction, formulate the main aim of the study based on gaps in existing knowledge.

4.        When constructing the model, what theory of deformation did the authors use? The theory of plastic flow with hardening or the deformation theory?

5.        Why did the authors choose the kinematic hardening model?

6.        The article does not have a section "Materials", only the section "Constitutive Model Theory" is given. At the same time, the application of deformation theories without describing the corresponding materials and their properties confuses the reader. I recommend adding.

7.        I recommend the authors to describe in more detail the process of numerical modeling and the features of the crystal plastic finite element method.

8.        The authors proposed a model. What is the accuracy of the model and what are the limits of its applicability?

9.        The "Results and Discussion" section does not contain a comparison of the results obtained by the authors with data from other researchers. I recommend that the authors expand the Discussion section, in which it is necessary to give an in-depth analysis of the results obtained and compare the results obtained by the authors with data from other researchers.

10.     Conclusions should be specified in terms of the scientific result and the scientific novelty obtained.

11.     English should be improved.

English should be improved.

Author Response

(The authors gave the same response as above.)

Reviewer 4 Report

Please see my detailed suggestions as listed below  

1.    Shorten the Abstract section.

2.    Adjust the aspect ratio of all figures. Label properly Figure 1.

3.    Polish your  introduction section by citing relevant studies.

4.    Please add more discussions on Figure 10 and highlight properly. What readers suppose to notice in these Figures ?

5.    “The void ratio of the nano-silver layer significantly impacts the joint's shear strength and thermal properties” what is the evidence of this claim ?

6.    Add more details in the results discussions section by expanding discussions on Figure 13 .

7.    See annotated file as well for improvements

8.    Improve conclusion section

9.    Check typo and spelling mistakes

Author Response

(The authors gave the same response as above.)
